# Alterations in the molecular composition of COVID-19 patient urine, detected using Raman spectroscopic/computational analysis

**John L. Robertson**[1,2,3‡*], **Ryan S. Senger**[3,4,5‡], **Janine Talty**[6‡], **Pang Du**[3,7�], **Amr Sayed-Issa**[3☍], **Maggie L. Avellar**[3,4☍], **Lacey T. Ngo**[3☍], **Mariana Gomez De La Espriella**[8☍], **Tasaduq N. Fazili**[8☍], **Jasmine Y. Jackson-Akers**[8☍], **Georgi Guruli**[9☍], **Giuseppe Orlando**[10☍]

**1** Department of Biomedical Engineering and Mechanics, College of Engineering, Virginia Tech, Blacksburg, Virginia, United States of America, **2** Section of Nephrology, Wake Forest University School of Medicine, Winston-Salem, North Carolina, United States of America, **3** DialySensors Incorporated, Blacksburg, Virginia, United States of America, **4** Department of Biological Systems Engineering, College of Life Sciences and Agriculture, Virginia Tech, Blacksburg, Virginia, United States of America, **5** Department of Chemical Engineering, College of Engineering, Virginia Tech, Blacksburg, Virginia, United States of America, **6** Clinical Biomechanics and Orthopedic Medicine, Roanoke, Virginia, United States of America, **7** Department of Statistics, College of Science, Virginia Tech, Blacksburg, Virginia, United States of America, **8** Internal Medicine/Infectious Disease, Carilion Clinic, Roanoke, Virginia, United States of America, **9** Division of Surgical Urology/Urologic Oncology, Department of Surgery, Virginia Commonwealth University, Richmond, Virginia, United States of America, **10** Department of Surgery, Wake Forest University School of Medicine, Winston-Salem, North Carolina, United States of America

☍ These authors contributed equally to this work.
‡ JLR, RSS and JT are joint senior authors on this work.
* drbob@vt.edu

**Data Availability Statement:** Data relevant to this study are available from Github at https://github.com/SengerLab/Raman-Scans/tree/COVID-19.

## Abstract

We developed and tested a method to detect COVID-19 disease, using urine specimens. The technology is based on Raman spectroscopy and computational analysis. It does not detect SARS-CoV-2 virus or viral components, but rather a urine 'molecular fingerprint', representing systemic metabolic, inflammatory, and immunologic reactions to infection. We analyzed voided urine specimens from 46 symptomatic COVID-19 patients with positive real time-polymerase chain reaction (RT-PCR) tests for infection or household contact with test-positive patients. We compared their urine Raman spectra with urine Raman spectra from healthy individuals (n = 185), peritoneal dialysis patients (n = 20), and patients with active bladder cancer (n = 17), collected between 2016–2018 (i.e., pre-COVID-19). We also compared all urine Raman spectra with urine specimens collected from healthy, fully vaccinated volunteers (n = 19) from July to September 2021. Disease severity (primarily respiratory) ranged among mild (n = 25), moderate (n = 14), and severe (n = 7). Seventy percent of patients sought evaluation within 14 days of onset. One severely affected patient was hospitalized, the remainder being managed with home/ambulatory care. Twenty patients had clinical pathology profiling. Seven of 20 patients had mildly elevated serum creatinine values (>0.9 mg/dl; range 0.9–1.34 mg/dl) and 6/7 of these patients also had estimated glomerular filtration rates (eGFR) <90 mL/min/1.73m$^2$ (range 59–84 mL/min/1.73m$^2$). We could not determine if any of these patients had antecedent clinical pathology abnormalities. Our

**Funding:** The authors received no specific funding for this work.

**Competing interests:** I have read the journal's policy and authors of this manuscript (JR, RS, PD) have the following competing interest. These authors have declared IP interests in the technology and Virginia Tech is seeking provisional patents for the technology that VT may intend to commercialize.

technology (Raman Chemometric Urinalysis—Rametrix®) had an overall prediction accuracy of 97.6% for detecting complex, multimolecular fingerprints in urine associated with COVID-19 disease. The sensitivity of this model for detecting COVID-19 was 90.9%. The specificity was 98.8%, the positive predictive value was 93.0%, and the negative predictive value was 98.4%. In assessing severity, the method showed to be accurate in identifying symptoms as mild, moderate, or severe (random chance = 33%) based on the urine multimolecular fingerprint. Finally, a fingerprint of 'Long COVID-19' symptoms (defined as lasting longer than 30 days) was located in urine. Our methods were able to locate the presence of this fingerprint with 70.0% sensitivity and 98.7% specificity in leave-one-out cross-validation analysis. Further validation testing will include sampling more patients, examining correlations of disease severity and/or duration, and employing metabolomic analysis (Gas Chromatography–Mass Spectrometry [GC-MS], High Performance Liquid Chromatography [HPLC]) to identify individual components contributing to COVID-19 molecular fingerprints.

## Introduction

Infection with SARS-CoV-2 and development of COVID-19 disease is associated with a deleterious effect in renal function and structure. This would be expected to potentially alter the molecular composition of urine. Since COVID-19 evolved in 2019, there have been numerous reports of acute kidney injury (AKI) associated with this infection [1–8]. The incidence of AKI in COVID-19 patients has been estimated to range from about 27–50+% [9, 10]. Early in the pandemic, several groups noted a correlation of disease severity, hospitalization, and intensive care admissions with increased risk for developing AKI [11, 12]. This was not surprising. The contribution of cardiopulmonary dysfunction, renal hypoperfusion, and/or multidrug therapy leading to the development AKI has been well-known for decades prior to COVID-19 [13]. However, the role of renal viral infection in the development of AKI was uncertain.

A recent review by Hassler and coworkers [14] considered evidence both for and against direct SARS-CoV-2 infection of the kidney. Rightly so, Hassler, et. al., and authors they cited, felt that viral infection might help explain the disproportionately high incidence of AKI and collapsing glomerulopathy seen in patients with COVID-19. Combining data from several studies, Hassler, et. al., presented at least putative evidence of viral infection in 102/235 kidneys (43%) from autopsied patients. A variety of techniques for viral/viral RNA detection were used in the cited studies, including immunohistochemistry, real-time polymerase chain reaction (RT-PCR), *in situ* hybridization, immunofluorescent microscopy, and electron microscopy [14]. No study they referenced used two or more methods for cross-checking and validating renal viral infection, a deficit in study design considered to affect interpretation in the results.

Hassler, et. al., [14] posited that renal biopsies (not autopsy-derived samples) and development of urine-based screening tests would be keys to understanding the effects of COVID-19 on renal function and structure and these would be needed to improve detection and management of disease.

Over the past four years, we have been developing and validating a novel approach to molecular urinalysis, using a combination of Raman spectroscopic, computational, and physicochemical analytical methods. This new method of urinalysis is termed Raman chemometric urinalysis—abbreviated as Rametrix®. We have successfully applied this method of urinalysis to determining the molecular characteristics and physical properties of urine specimens from

healthy human volunteers (n = 235 urine samples) [15], patients with chronic kidney disease (CKD) (n = 362 urine samples) [16], bladder cancer (BCa) and non-neoplastic genitourinary pathologies (such as cystitis and benign prostatic hypertrophy/urinary retention, among others) (n = 56 urine samples) [17, 18], and post-acute patients following exposure to Lyme pathogen (*Borrelia* sp.) (n = 30 urine samples) [19]. Data cited in these references was derived from Rametrix® analysis of urine specimens collected prior to December 2018.

Normal human urine contains over 2,000 separate chemical entities, reflective of systemic physiology/metabolism and the processes of renal function [20]. In the past decade, mass spectrometry, liquid/gas chromatography, nuclear magnetic resonance, and kinetic nephelometry methods have been used to detect analytes (i.e., biomarkers) associated with normal metabolism or disease [21]. Metabolomics has been used to identify kidney disorders (including renal cell carcinoma), coronary artery disease, diabetes, Alzheimer's disease and cognitive impairment, neurodegenerative disease, and colorectal cancer [22–27]. While metabolomics is often used to search for circulating/plasma disease biomarkers, it is now used to study how the presence of disease alters the urine metabolite profile ("fingerprint").

Mass spectrometry-based urine biomarker and "-omics" technologies are used rarely by caregivers in patient care settings. This is due to expense, the daunting requirement for advanced technology, expertise required for interpretation of results, and the lack of assay validation requiring large datasets of normal and abnormal specimens. In fact, the complexity of both acute and chronic genitourinary tract pathologies makes large dataset sampling and validation with technology-intensive methods (like mass spectrometry and high-performance liquid chromatography) unlikely and cost-prohibitive.

As an alternative approach to mass spectroscopy-based urine metabolomics, we invented and extensively validated Rametrix® to analyze urine [15, 28, 29]. Raman spectroscopy is a mature, well-studied, and powerful technology that has been applied to analysis of the chemical composition of a wide variety of solids and liquids, including biological specimens [30–33]. Irradiation of molecular mixtures (like urine), with wavelength-specific laser energy, produces weak vibrational energy (Raman scatter radiation) from deformation/relaxation of the many chemical bonds in hundreds of distinct molecules in specimens. Different molecular constituents are represented by Raman 'bands' (i.e., signal intensity peaks) and these bands/peaks are indicative of chemical bond vibrations [34]. These vibrations may be present in several molecules with similar chemical bonds in a sample, meaning it can be difficult to assign individual Raman bands to specific molecules, unless they are present in abundance. This is the case for urea in urine, for example, where the C-N bond stretch at 1,002 cm$^{-1}$ is dominant and can be associated with urea concentration. We have also identified the bands/peaks of creatinine, heme, amino acids, albumin, collagen, and phospholipids in urine. A few of these and other broad molecular assignments are shown in Raman spectra of urine from healthy volunteers [15], CKD 4–5 patients [16], and Surine™ urinalysis analytical control solution (Dyna-Tek Industries, Lenexa, KS) in Fig 1.

Because it is difficult to relate individual Raman bands to specific molecules, a chemometric approach is required to analyze Raman spectra of highly complex heterogenous samples [28, 35–39]. The chemometric approach is unlike chromatographic and mass spectrometry approaches that resolve single molecules. The chemometric approach treats an entire Raman spectrum as a 'fingerprint' and then associates it with a condition (i.e., 'healthy', 'chronic kidney disease,' 'COVID-19 infection', etc.) using statistical models and artificial intelligence. Building an accurate model to predict the condition of an unknown sample requires a large dataset of pre-analyzed Raman spectra. This can be seen, for example, in Fig 1. Here, representative urine spectra are shown for healthy volunteers and patients with diagnosed disease. Chemometric models determine whether an "unknown" patient sample more closely resembles

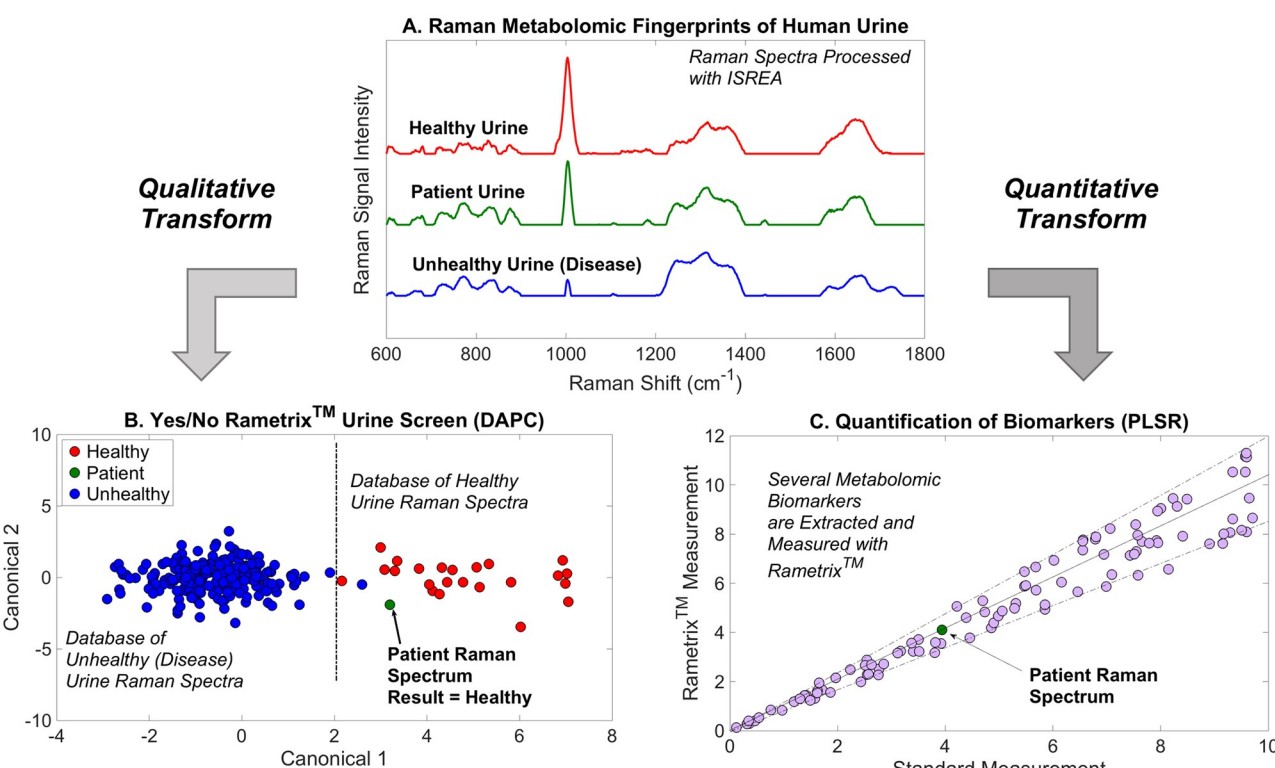

**Fig 1. Schematic of qualitative and quantitative chemometric analyses of urine spectra using the Rametrix® Toolbox.**

the healthy urine spectrum or one of a diseased state, without knowing the identities of all molecules in each sample.

Rametrix® computations are performed using the Rametrix® Toolbox for MATLAB, which is available to academic researchers through GitHub [28, 29]. The Rametrix® Toolbox offers two approaches to data analysis (also shown in Fig 1), (i) qualitative classification, and (ii) quantitative analysis. For classification (e.g., "yes/no" to the presence of disease) the Rametrix® Toolbox offers principal component analysis (PCA) followed by discriminant analysis of principal components (DAPC) [28, 29, 32]. Other deep learning and artificial neural network options are available, and other classifier models used with Raman spectra were surveyed recently [32]. Partial least-squares regression (PLSR) for quantitative analyses, and methods to identify quantifiable biomarkers from Raman spectra have been produced recently [17]. The Rametrix® Toolbox also includes ISREA [40, 41], which enables us to "exclude" non-diagnostic features in samples/spectra, such as the presence of blood/breakdown products, if such exclusion is warranted and logical [17, 32]. Finally, the Rametrix® Toolbox offers a graphical interface, spectral viewing pane, and predictive model cross-validation through leave-one-out analysis [29].

Analysis of urine by Rametrix® is inexpensive (uses off-the-shelf Raman spectrometers and costs a few dollars a sample for consumables and analysis), rapid (typically less than 30 minutes to process and interpret a sample), requires no urine sample preparation or chemical manipulation, and is non-invasive (voided samples are analyzed).

These characteristics, and ease-of-use, suggested to us that this Raman spectroscopy-based technology might be useful in determining if a COVID-19 'molecular fingerprint' was present

in the urine of diseased patients. If so, this fingerprint could be useful in disease detection and patient management.

## Methods

### Patients and controls

**Informed consent.**   The collection and analysis of all de-identified urine specimens reported in this study and Raman spectra from previously reported reference studies were performed in accordance with the principles of the Declaration of Helsinki. Informed consent for the collection of urine specimens was obtained from healthy individuals (no evidence of COVID-19 disease) and RT-PCR-positive COVID-19 patients. Following the description of the proposed study by study personnel to prospective participants, patients who provided verbal consent for the study were provided with specimen cups, urine collected, and the specimen was encoded by study personnel. Informed consent for participation was recorded in patient medical records securely retained by study personnel (JT). This study, and previously reported studies cited in this document, were approved by the Virginia Tech Institutional Review Board (VT IRB #s 15–703, 20–924, 21–569) and Virginia Commonwealth University Institutional Review Board (reference bladder cancer study) (VCU IRB # HM20006879).

**Study subjects: COVID-19 patients.**   Forty-six (46) patients, with clinical signs of COVID-19 disease, RT-PCR confirmation of nasopharyngeal infection and/or household/congregate and temporal exposure to RT-PCR confirmed patients, were seen by a primary care physician for disease/symptom management. All patients were symptomatic, but ambulatory, at the time of evaluation. The patient population consisted of 32 female and 14 male patients. The age range of female patients was 18–68 years (average age 47.84 years/old) and of male patients was 18–62 years (average age 47.85 years/old). As would be expected, the clinical presentation of patients was highly variable, as was the duration and severity of clinical symptoms. Thirty out of 46 patients (30/46) were seen for evaluation (and specimen collection) within the first 14 days of clinical disease. Twelve of 46 patients (12/46) had clinical disease present for 30–300 days, and 10 of 46 patients (10/46) had clinical disease present for 60–300 days. Based on physician evaluation and self-assessment, 25/46 patients presented with 'mild, symptomatic' disease, while 14/46 patients presented with 'moderate, symptomatic' disease. Seven of 46 patients (7/46) were classified 'severe, symptomatic' at the time of presentation and specimen collection.

Several patients had pursued multiple avenues of diagnosis and variable courses and types of therapies (including antibiotics) prior to evaluation.

Clinical pathology evaluation (serum chemistry) was performed on a subset of 20/33 patients. Seven of 20 patients had mildly elevated serum creatinine values (>0.9 mg/dl; range 0.9–1.34 mg/dl) and 6/7 of these patients also had eGFR <90 mL/min/1.73m$^2$ (range 59–84 mL/min/1.73m$^2$).

**Controls: Healthy volunteers (pre-COVID-19).**   A full analysis of the healthy human volunteer urine dataset has been published [15]. This dataset contains 235 urine specimens collected from 39 females and 9 males, and all were collected prior to December of 2018. All volunteers were healthy (free of infectious or degenerative disease) at the time of collection and had no history or evidence of renal disease. The population ranged in age from 18 to 70 years, and 87.5% were between 19–22 years of age (median of 21 years). A total of 185 urine spectra were selected randomly from this dataset and used in this study.

**Controls: Healthy volunteers (fully vaccinated against COVID-19).**   For this study, 19 additional urine specimens were collected from healthy volunteers who had been fully

vaccinated against COVID-19 and who had no history or evidence of either renal disease or COVID-19 disease.

**Controls: End-stage renal disease (ESRD) patients (pre-COVID-19).** The ESRD patient urine dataset has also been published [16]. It contains 362 urine specimens from 96 patients receiving treatment for ESRD with peritoneal dialysis therapy. The age range was 24 to 90 years, with a mean of 60 and median of 63.5 years. Twenty (20) spectra were selected randomly from this dataset and used in this study.

**Controls: Bladder cancer (BCa) patients (pre-COVID-19).** We have also published urine spectra from patients with active or remissive BCa [18]. The dataset contains 56 urine specimens (one per patient) from patients between 31–91 years old (mean and median of 62 years) were collected. The patients ranged in age from 31–91 years old. The mean and median age of 62 years. From this dataset, we selected 17 specimens from patients with active BCa at the time of collection for this study.

**Specimen collection and storage.** Voided, mid-stream urine specimens were collected, frozen immediately at -15°C, and stored at -35°C until analysis. We have demonstrated the suitability of this procedure for preserving samples [18].

**Raman methodology and measurements.** Urine specimens were analyzed at room temperature in bulk liquid form using 2 mL screw thread flat bottom borosilicate glass vials (Fisher Scientific). A Wasatch Photonics 785 nm dispersive Raman spectrometer (Wasatch Photonics, Morrisville, NC) was used with a Rametrix® AutoScanner (DialySensors, Inc., Blacksburg, VA) to automate sample scanning. The following settings were used: 25°C, 785 nm laser, 30 s excitation time, 30 mW laser power, 0.2 mm laser spot size, 200–2000 cm$^{-1}$ range, and spectral resolution of 8 cm$^{-1}$ (manufacturer default). Ten scans were obtained per vial. ENLIGHTEN™ software (Wasatch Photonics) was used for spectrometer operation, and molecular contributions investigated with a published database [42]. In all cases, Raman intensity and wavenumber calibrations were performed during each operation of the Raman spectrometer using Surine™ urine analytical control (see below) and published chemometric protocols.

**Analytical standards.** Surine™ Urine Negative Control (Dyna-Tek Industries, Lenexa, KS) was used as a control in this study.

**Computational methodology.** Previously published computational methods were used [19, 28, 29, 42] with the Rametrix® Toolbox (LITE v1.1 and PRO v1.0) with added capabilities for ISREA baselining [40, 41]. Calculations were performed in MATLAB R2018A (Mathworks; Natick, MA). Raman spectra were truncated to 600–1800 cm$^{-1}$, baseline corrected with ISREA, averaged over the 10 scans for each urine specimen, and vector normalized. ISREA was applied using nodes (or 'knots') at wavenumbers of 400, 950, 1100,1500, and 1800 cm$^{-1}$. In specified cases, the placement of nodes was also allowed to vary, as described previously [19], to exclude specific Raman shift regions of spectra selectively. Spectra were analyzed by principal component analysis (PCA) and discriminant analysis of principal components (DAPC) in the Rametrix® LITE Toolbox, and models were cross-validated with leave-one-out analysis with Rametrix® PRO. These methods have been implemented and described previously [19, 28, 32, 35–39]. This procedure allowed calculation of overall prediction accuracy, sensitivity, specificity, positive-predictive value (PPV), and negative-predictive value (NPV) for detecting COVID-19 in human urine. These metrics have also been defined in previous publications, and here, the RT-PCR nasopharyngeal swab test and/or proximate/congregate/temporal exposure to COVID-19 positive patients is treated as the "Gold-Standard" test when comparing to the Rametrix® urine screen.

**Statistical comparisons.** Of spectra were performed through the calculation of total spectral distance (TSD), as has been demonstrated [15, 16, 18, 19, 29, 32]. In calculation of TSD, the difference between each urine spectrum and that of Surine™ was calculated at each

wavenumber and summed. One-Way Analysis-of-Variance (ANOVA) and pairwise comparisons using Tukey's honestly significant difference (HSD) procedure were used to determine if TSD values of COVID-19 patient urine were different from those of healthy volunteers and those with other diseases.

## Results

### Raman dataset

Urine Raman spectra from the patient dataset shown in Table 1 were used in this study. Spectra obtained from patients with active COVID-19 are a new contribution in this study, along with spectra obtained from healthy volunteers vaccinated for COVID-19 but with no known history or exposure to COVID-19. Other Raman scans for healthy human volunteers (collected and analyzed pre-2019; i.e., before COVID-19).

### Spectral processing

All spectra were truncated between 600–1,800 cm$^{-1}$, baselined using ISREA [40, 41], vector normalized, and averaged for each urine specimen. For the ISREA implementation, nodes were applied at 400, 950, 1100, 1500, and 1800 cm$^{-1}$. The concept of ISREA node placement and optimization has also been introduced recently [32]. Averaged spectra from all classes listed in Table 1 are shown in Fig 2. The most notable observable difference from the spectra of the COVID-19 and Healthy (pre-2019) classes was the height of the urea representative band (1,002 cm$^{-1}$). Inspection also revealed other minor differences (e.g., 970 cm$^{-1}$; 1100–1200 cm$^{-1}$), prompting additional differences to be investigated by chemometric methods. To date, patients with ESRD have the most visually different urine Raman spectra from the Healthy class (Fig 2). However, it is noted that the ESRD class is also identified by a reduced urea Raman band intensity.

### Statistical significance

Total Principal Component Distance (TPD) [18, 28, 29] calculations were performed to determine if urine Raman spectra of the "COVID-19" class were different from all other classes (combined to form a "non-COVID-19" class). In short, TPD uses ISREA baselined and

**Table 1. Patient dataset analyzed in this study.**

| Number of Urine Specimens | Description | Classification | Reference |
|---|---|---|---|
| 185 | Healthy human volunteers (pre-2019) | "Healthy" | Senger et al. 2019 |
| 20 | Peritoneal dialysis patients with CKD 4–5 | "ESRD" | Senger et al. 2020 |
| 17 | Patients with active bladder cancer | "BCa" | Huttanus et al. 2020 |
| 6 | Surine™ (lot from 2016) | "Surine" | Huttanus et al. 2020 and This study |
| 5 | Surine™ (lot from 2021) | "Surine" | This study |
| 19 | Healthy human COVID-19 vaccinated volunteers (2021) | "Healthy" | This study |
| 46 | Patients with active COVID-19 | "COVID-19" | This study |
| 25 | Patients with 'mild' severity COVID-19 symptoms | "COVID-19 (Mild)" | This study |
| 14 | Patients with 'moderate' severity COVID-19 symptoms | "COVID-19 (Moderate)" | This study |
| 7 | Patients with 'severe' COVID-19 symptoms | "COVID-19 (Severe)" | This study |
| 12 | Patients with COVID-19 clinical disease lasting longer than 30 days | "COVID-19 (Long COVID 19)" | This study |

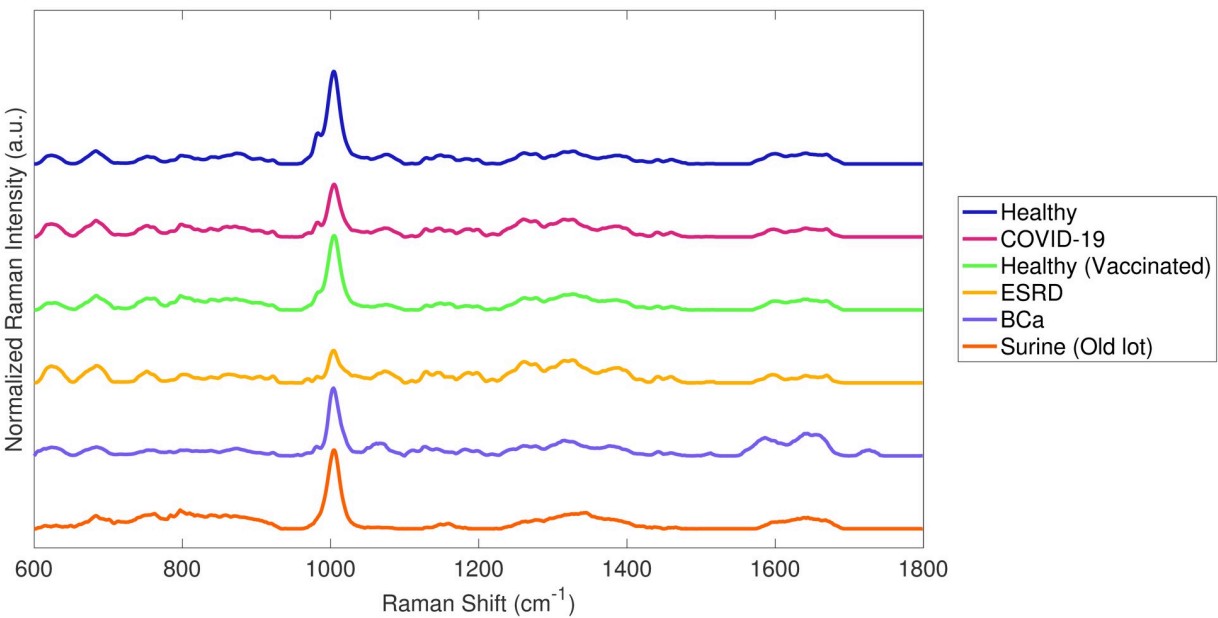

**Fig 2. Averages of ISREA baselined and vector normalized spectra for classes specified in Table 1.**

processed spectra. The first five principal components (PCs) of PCA are used. For each sample, the distance (across all five PCs) is calculated between that sample and Surine™ (using a simple distance formula). This provides a distance calculation for every sample in the dataset. Then, ANOVA and pairwise comparisons are used to determine if statistically significant distances exist between classes of spectra. TPD was applied to the COVID-19 class and all other classes grouped as non-COVID-19. Through this method, the COVID-19 class and non-COVID-19 grouped class were found statistically significant ($p < 0.001$). Pairwise comparisons were applied, and the COVID-19 class was found statistically different ($p < 0.001$) against all other groups in Table 1. From experience, this indicates that an effective predictive model may be able to be constructed from PCA followed by DAPC. Of other pairwise comparisons, it was found that the Healthy and Surine™ groups were not statistically different from one another ($p = 0.79$) according to TPD calculations.

## Predictive urine screen for COVID-19

With the COVID-19 class showing statistical significance from all other classes, predictive models were built using PCA and DAPC. The model inputs were truncated, baselined, and normalized urine spectra, and the ISREA nodes of 400, 950, 1100, 1500, and 1800 cm$^{-1}$ were used in this initial model-building. Spectra were processed further by PCA to produce principal components (PCs). A specified number of PCs were then fed into DAPC to return a "yes/no" for the presence of COVID-19. Predictive models differed by the number of PCs fed into DAPC, and these were evaluated for performance by leave-one-out cross-validation. Results are shown in Fig 3A for a model designed to separate all classes in Table 1 using 99% of the dataset variance (available in the top 20 PCs). This plot was effective in showing cluster separation, particularly that the COVID-19 cluster separated from the Healthy cluster more effectively than the BCa group did. The separation of COVID-19 and the non-COVID-19 groups for this model is shown in Fig 3B. When cross-validated with leave-one-out, an overall prediction accuracy of 97.6% for our dataset. The sensitivity of this model for detecting COVID-19

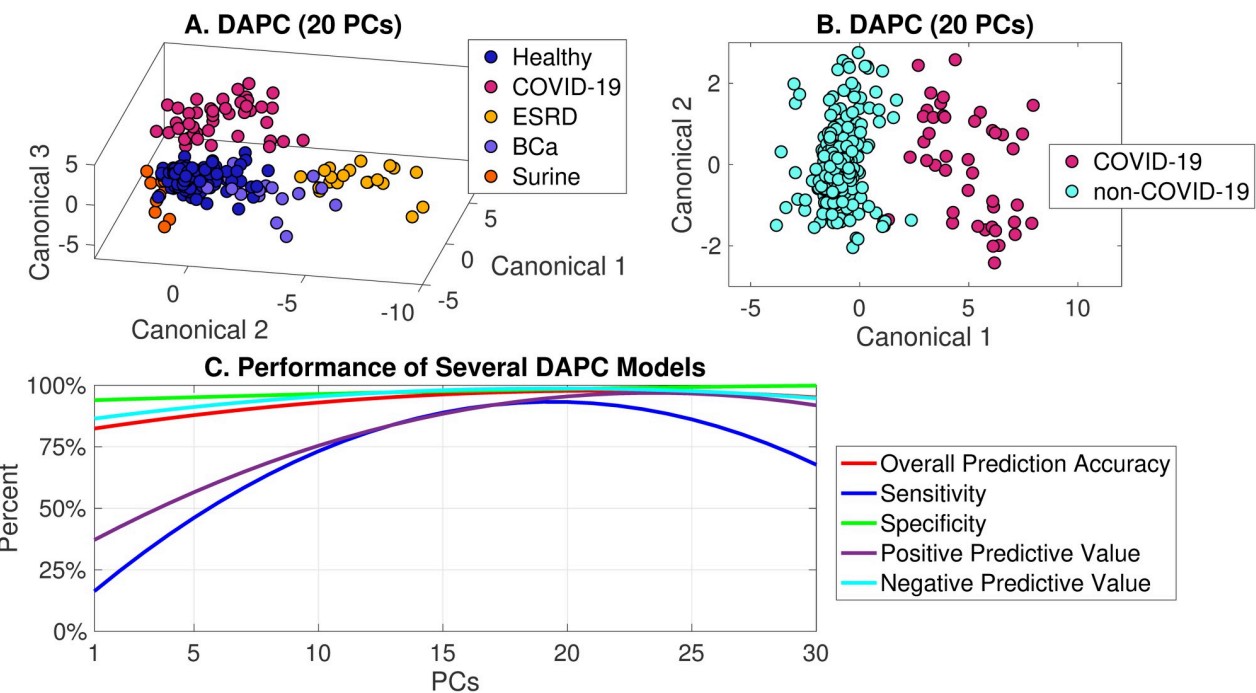

**Fig 3. DAPC models demonstrating cluster separation of COVID-19 Raman urine spectra from those of other groups (A, B), and DAPC predictive model performance (C).**

was 90.9%. The specificity was 98.8%, the positive predictive value (PPV) was 93.0%, and the negative predictive value (NPV) was 95.8%. Finally, the performance of several DAPC models is given in Fig 3C and illustrate the influence of model architecture on performance. It is noted that at least 6 PCs were required to obtain overall prediction accuracy, sensitivity, specificity, positive predictive value, and negative predictive value above 50%.

Next, we varied the location of the ISREA nodes with the objective of reducing the number of PCs required in DAPC. Several iterations were tried, and node positions of 400, 439, 446, 605, 1045, 1163, 1247, 1443, 1739, 1768, and 1775 cm$^{-1}$ yielded similar results to those reported above; however, only 4 PCs of the dataset were required (as opposed to 20). A comparison of the two node sets for detecting the presence of COVID-19 in urine with Rametrix$^{®}$ analysis is shown in Table 2.

## Molecular contributions

The separation of COVID-19 and non-COVID-19 cluster separation in Fig 3B was investigated further through PC and canonical (DAPC) loadings. Significant Raman shifts (defined as above 0.2% total contribution) are given in Table 3. The molecular assignments were obtained by a published database [43]. It was observed the occurrences of lipids/cholesterol,

**Table 2. Detection of COVID-19 in urine by Rametrix$^{®}$ given two different ISREA node sets.**

| ISREA Nodes | PCs | Overall Accuracy | Sensitivity | Specificity | PPV | NPV |
|---|---|---|---|---|---|---|
| 400, 950, 1100,1500, and 1800 cm$^{-1}$ | 20 | 97.6% | 90.9% | 98.8% | 93.0% | 95.8% |
| 400, 439, 446, 605, 1045, 1163, 1247, 1443, 1739, 1768, and 1775 cm$^{-1}$ | 4 | 97.6% | 93.2% | 98.4% | 91.1% | 98.8% |

**Table 3. Molecular assignments for Raman shifts leading to cluster separations in Fig 3.**

| Raman Shift (cm$^{-1}$) | Present in PCA, DAPC, or Both | Molecular Assignment [43] |
|---|---|---|
| 425 | Both | N/A |
| 445 | Both | N-C-S stretch |
| 485 | Both | Glycogen |
| 518 | DAPC | Phosphatidylinositol |
| 614 | DAPC | Cholesterol ester |
| 621 | Both | C-C twisting of phenylalanine |
| 627 | DAPC | N/A |
| 682 | DAPC | N/A |
| 688 | PCA | N/A |
| 702 | DAPC | Cholesterol ester |
| 719 | PCA | Lipids |
| 776 | PCA | Phosphatidyl inositol |
| 782 | PCA | DNA |
| 810 | PCA | Phosphodiester |
| 817 | PCA | Collagen |
| 830 | PCA | Phosphate stretching, Tyrosine |
| 847 | PCA | Monosaccharides |
| 860 | DAPC | Phosphate group |
| 880 | Both | Tryptophan |
| 893 | PCA | C-C backbone |
| 900 | DAPC | N/A |
| 906 | DAPC | Tyrosine |
| 913 | DAPC | Glucose |
| 955 | PCA | Carotenoids |
| 980 | Both | Beta-sheet proteins |
| 992 | DAPC | Red blood cell, phenylalanine, NADH |
| 1002 | Both | Urea |
| 1006 | Both | Carotenoids (absent in normal tissue) |
| 1008 | DAPC | Phenylalanine |
| 1013 | DAPC | N/A |
| 1030 | DAPC | Phenylalanine of collagen |
| 1049 | DAPC | Glycogen |
| 1058 | PCA | Lipids |
| 1073 | PCA | Fatty acids |
| 1077 | DAPC | Lipids, phospholipids, phosphate |
| 1080 | DAPC | Phospholipids, phosphate, collagen, tryptophan |
| 1104 | PCA | Phenylalanine |
| 1107 | PCA | N/A |
| 1126 | Both | Protein, disaccharides, lipids |
| 1185 | PCA | Phosphate |
| 1240 | DAPC | RNA, phosphate, collagen |
| 1327 | Both | Nucleic acids |
| 1396 | Both | Beta-carotene |
| 1491 | PCA | Amino radical cations |
| 1607 | Both | Tyrosine and phenylalanine |
| 1630 | PCA | N/A |
| 1641 | DAPC | N/A |

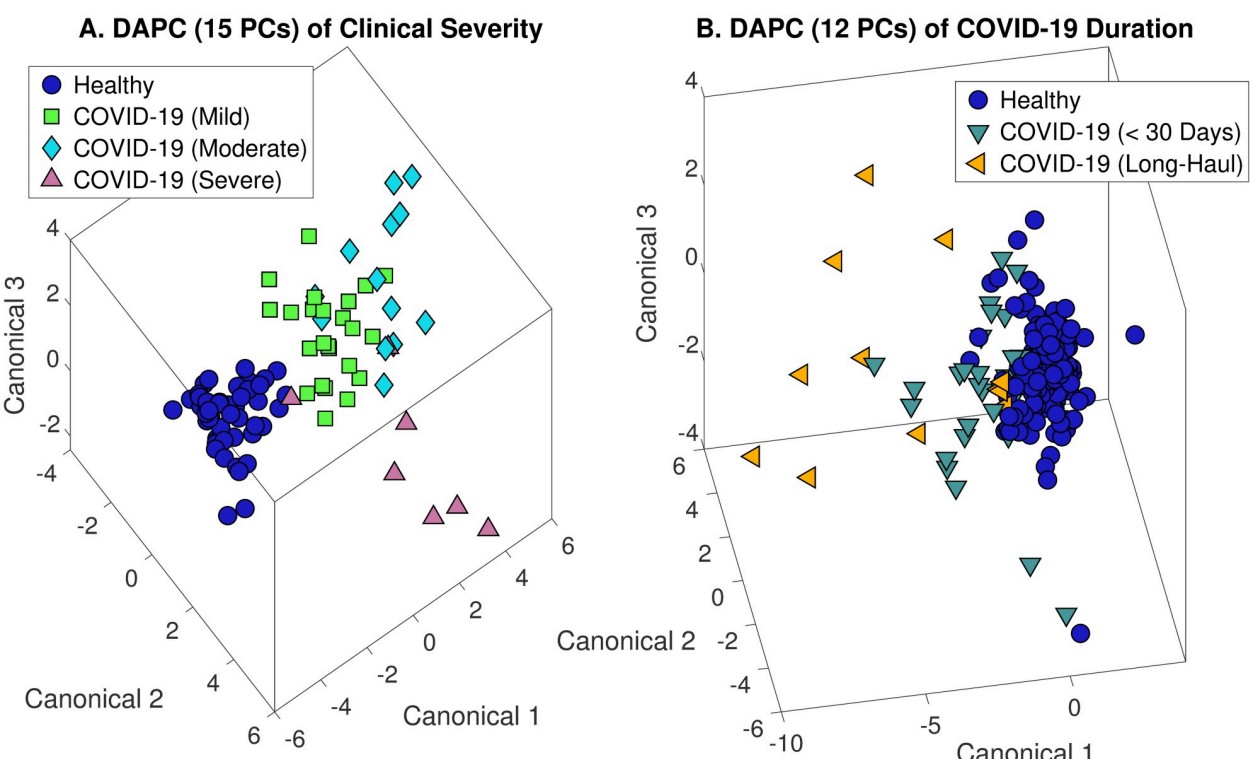

**Fig 4. DAPC models demonstrating cluster separation of COVID-19 Raman urine spectra by clinical severity (A) and duration of symptoms (Long-Haul) (B).**

collagen, and phosphates occurred more regularly than our similar analyses of CKD and BCa [16, 18].

## Severity of clinical symptoms

The separation of COVID-19 urine Raman scans by severity of clinical symptoms from those of healthy volunteers is shown in Fig 4A. When clustered by DAPC, the COVID-19 data largely separated by severity, with mild symptoms clustering closer to healthy scans, on average. However, we note one urine sample from a COVID-19 patient with severe symptoms clustered with the healthy group. This is unexplained by our analysis. With separation by severity, the presence of COVID-19 was detected with 93.5% overall accuracy (87.5% sensitivity, 100% specificity, 100% PPV, and 88.0% NPV). When inspecting among COVID-19 samples to determine severity, 60–66% overall accuracy was achieved for the three levels (mild, moderate, and severe). Full results are given in Table 4. We note that given the three levels of severity, the random chance of correct prediction is 33%.

**Table 4. Prediction of COVID-19 clinical severity by Rametrix® analysis of urine.**

| Clinical Severity* | Accuracy | Sensitivity | Specificity | PPV | NPV |
|---|---|---|---|---|---|
| Mild | 59.1% | 79.2% | 35% | 59.4% | 58.3% |
| Moderate | 65.9% | 61.5% | 67.7% | 44.4% | 80.8% |
| Severe | 65.9% | 57.1% | 67.6% | 25% | 89.3% |

* Random chance of correct prediction is 33%.

**Table 5. Detection of long COVID-19 in urine by Rametrix® given two different ISREA node sets.**

| ISREA Nodes | PCs | Overall Accuracy* | Sensitivity | Specificity | PPV | NPV |
|---|---|---|---|---|---|---|
| 400, 950, 1100,1500, and 1800 cm⁻¹ | 12 | 95.1% | 25.0% | 98.7% | 50.0% | 96.2% |
| 600, 1074, 1153, 1230, 1313, 1416, 1507, 1800 cm⁻¹ | 4 | 97.6% | 70.0% | 98.7% | 70.0% | 98.7% |

* Random chance of correct prediction is 50%.

### Identification of long COVID19

Of the patients treated for COVID-19, 26% (12/46) showed symptoms longer than 30 days. Of this group, over 83% showed symptoms for longer than 60 days (up to 300 days). Thus, this group of patients was defined as 'COVID-19 (Long COVID19)'. A molecular fingerprint was sought to determine if urine metabolomic differences existed between the Long COVID19 group and those whose symptoms resolved in less than 30 days (regardless of clinical severity). Using the initial ISREA node set (400, 950, 1100,1500, and 1800 cm⁻¹), no significance was found, where prediction accuracy, sensitivity, specificity, PPV, and NPV all exceeded 50% (the random chance of correct prediction). To find a signal, we shortened the truncation range to 600–1800 cm⁻¹ and searched for additional ISREA nodes. Ultimately, a node set was located that led to 70% sensitivity, with better than 98% specificity, for detecting Long-Haul COVID-19. The results are shown in Fig 4B, and a comparison of the ISREA node sets with prediction metrics is given in Table 5. We note that the plot in Fig 4B was constructed with 12 PCs (instead of 4) to better show the separation of Long-Haul samples.

## Discussion

We believe this is the first study demonstrating that SARS-CoV-2 infection changes the chemical composition of urine. These changes–complex, multimolecular 'fingerprints'—can be detected using Raman spectroscopic examination and computational analysis. Sample analysis is low-cost (dollars per sample) and rapid (results <30 minutes). This analytical method does not detect virus or viral components. This method also does not identify a single "biomarker" of COVID-19 disease, but rather a "biomarker pattern" composed of molecular clusters associated with disease. These biomarker patterns reflect systemic inflammatory, immunologic, and metabolic reactions to infection. We hypothesize that viral infection of the kidney (if substantiated) may affect renal form/function and urine composition. Our results support information in many of the studies reviewed/critiqued by Hassler, et. al. [14].

This study would not have been possible without access to a large database of urine Raman spectra (n = 235) from healthy volunteers collected prior to SARS-CoV-2/COVID-19 and other large databases of urine Raman spectra (pre-COVID-19 CKD and bladder cancer patients) These provided the critical context for interpretation of spectra from COVID-19 patients. Without access to large, pre-COVID-19 spectral databases, we could not have been sure that what was observed in our study was related to SARS-CoV-2 infection. Other investigators may find that a lack of pre-COVID-19 clinical samples (urine, serum, from healthy and diseased individuals, for example) is a challenge as they search for COVID-19-related biomarkers.

Our understanding of the myriad effects of COVID-19 disease is in its infancy. The tropisms of the pathogen, clinical course of infection, the ongoing evolution of variants under immunologic pressure, and individual responses to infection have only been intensively studied for slightly more than 23 months. Largely unknown are long-term effects of infection. We do not know if acute infection evolves to chronic disease and organ dysfunction. This may be

especially important for COVID-19 patients who had antecedent renal disease. By analogy, Yu and Bonventre [44] noted the complex interplay between diabetic kidney disease (DKD), diabetes mellitus (DM), and AKI. They noted that DKD/DM patients were more likely to develop AKI and that the interactions of these disease process affected development of CKD and ESRD. One might expect that patients with COVID-19-associated AKI might be at increased risk for progression of chronic renal disease.

We acknowledge limitations of this study. The small number of patients studied (n = 46) limits computational comparisons and correlations of disease severity and duration. These comparisons will require much more sampling and data analysis. None of the patients studied had AKI, although standard laboratory metrics (serum creatinine, eGFR) for patients indicated a mild degree of renal dysfunction. We did not follow patients with serial samples and therefore it is not possible to determine if there is a progression of renal dysfunction or a return to pre-COVID renal metrics. These limitations will be addressed in ongoing and planned studies.

What are the next steps? First, we need to conduct a larger study of COVID-19 patients to determine 1) are the results of this preliminary study validated with further sampling, 2) how does the molecular fingerprint vary among individuals, 3) does the molecular fingerprint differ in individuals with different disease severities and durations, 4) how long does the molecular fingerprint persist following acute infection, 5) can the molecular fingerprint specifically indicate renal infection, and 6) can the molecular fingerprint predict development of AKI? Second, the important molecular clusters contributing to the molecular fingerprint will need to be studied with confirmatory metabolomic analysis. With this additional information, we may have a validated, non-invasive, inexpensive method to monitor systemic manifestations of disease. This could be used to detect infections potentially missed with current PCR/antigen-based technologies and to monitor the efficacy of therapy and/or detect possible disease progression. We expect it will be a very useful tool for monitoring direct/indirect renal effects of COVID-19 disease. This technology, once more fully validated, could easily be used for non-invasive, repetitive monitoring of individuals who choose not to be vaccinated, to detect 'break-through' infections in vaccinated individuals, and to differentiate COVID-19 disease from seasonal respiratory infections (influenza), and may be especially useful in the detection and management of Long COVID19.

## Conclusions

Our preliminary data shows that SARS-CoV-2 infection and COVID-19 disease alters the molecular composition of urine, as determined by Raman spectroscopy and computational analysis. We believe our findings could be applied to disease detection, early screening for serious renal complications such as AKI, and overall management of COVID-19.

## Author Contributions

**Conceptualization:** John L. Robertson, Ryan S. Senger, Mariana Gomez De La Espriella, Georgi Guruli.

**Data curation:** John L. Robertson, Ryan S. Senger, Janine Talty, Pang Du, Amr Sayed-Issa, Lacey T. Ngo, Mariana Gomez De La Espriella, Jasmine Y. Jackson-Akers, Georgi Guruli.

**Formal analysis:** John L. Robertson, Ryan S. Senger, Pang Du, Amr Sayed-Issa, Maggie L. Avellar, Lacey T. Ngo.

**Investigation:** John L. Robertson, Ryan S. Senger, Amr Sayed-Issa, Mariana Gomez De La Espriella, Tasaduq N. Fazili, Jasmine Y. Jackson-Akers, Georgi Guruli, Giuseppe Orlando.

**Methodology:** John L. Robertson, Ryan S. Senger, Pang Du, Amr Sayed-Issa, Maggie L. Avellar, Lacey T. Ngo, Mariana Gomez De La Espriella, Georgi Guruli, Giuseppe Orlando.

**Project administration:** John L. Robertson.

**Resources:** John L. Robertson.

**Software:** Ryan S. Senger, Pang Du.

**Supervision:** John L. Robertson.

**Validation:** John L. Robertson, Ryan S. Senger.

**Visualization:** John L. Robertson, Ryan S. Senger.

**Writing – original draft:** John L. Robertson, Ryan S. Senger, Janine Talty, Pang Du, Maggie L. Avellar, Mariana Gomez De La Espriella, Tasaduq N. Fazili, Jasmine Y. Jackson-Akers, Georgi Guruli, Giuseppe Orlando.

**Writing – review & editing:** John L. Robertson, Ryan S. Senger, Mariana Gomez De La Espriella, Tasaduq N. Fazili, Jasmine Y. Jackson-Akers, Georgi Guruli, Giuseppe Orlando.

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
