## [Decision Letter · Decision Letter 0]

21 Feb 2022

PONE-D-21-38432Alterations in the molecular composition of COVID-19 patient urine, detected using Raman spectroscopic/computational analysisPLOS ONE

Dear Dr. Robertson,

Thank you for submitting your manuscript to PLOS ONE. After careful consideration, we feel that it has merit but does not fully meet PLOS ONE’s publication criteria as it currently stands. Therefore, we invite you to submit a revised version of the manuscript that addresses the minor points raised during the review process.

We look forward to receiving your revised manuscript.

Kind regards,

Benedetta Bussolati, MD, PhD

Academic Editor

PLOS ONE

Journal Requirements:

Reviewers' comments:

Reviewer's Responses to Questions

**Comments to the Author**

1. Is the manuscript technically sound, and do the data support the conclusions?

Reviewer #1: Partly

Reviewer #2: Yes

2. Has the statistical analysis been performed appropriately and rigorously? 

Reviewer #1: Yes

Reviewer #2: Yes

3. Have the authors made all data underlying the findings in their manuscript fully available?

Reviewer #1: Yes

Reviewer #2: Yes

4. Is the manuscript presented in an intelligible fashion and written in standard English?

Reviewer #1: Yes

Reviewer #2: Yes

5. Review Comments to the Author

Reviewer #1: This paper focuses on the effect of Covid-19 infection on the kidney. The research question raised in this manuscript by JL Robertson et al is whether chemometric analysis of urine using Raman spectroscopy data fed into a sophisticated computational and graphing instrument (Rametrix) can differentiate patients diagnosed with COVID-19 from both healthy subjects and groups of patients with other conditions (end-stage renal disease and bladder cancer).

The investigators have published extensively on the spectral analysis of biologic fluids (e.g., references 15-19,28,32,35-41). They clearly have the experience with Raman spectroscopic technology and computational expertise to allow for confidence in their technical expertise and interpretation of data. In this report they convincingly demonstrate that the spectral pattern derived from the urine of study subjects is distinct from those seen in the above groups of non-Covid subjects. The investigators provide compelling data on a timely subject that could lead to a rapid, inexpensive way to diagnose and monitor Covid-19 activity.

Major Issues

1. Although the authors acknowledge that the biomarker pattern that they observed in the urine of patients with Covid-19 are reflective of the 'systemic inflammatory, immunologic, and metabolic reactions to infection,’ they state that "the technology might differentiate COVID-19 disease from seasonal respiratory infections (influenza)". Since it may be the case that other infectious agents result in a similar urinary spectral 'fingerprint,' the latter inference should await validation from further studies incorporating patients with other infections. In this regard, there is a reference to work they did with Lyme disease that is currently in press (reference 19).

2a. It would be illuminating to know the selection criteria and explicit manner in which the authors recruited the 33 patients with COVID-19 disease. This diagnosis remains in some doubt for some number of them without further clarification as some of them are described in Methods as ‘household/congregate and temporal exposure to RT-PCR confirmed patients.’ If these individuals had close contact with index patients and had typical COVID-19 symptoms, however, that would justify their assignment to the group of study subjects.

2b. The sample of patients (n=33) with presumed COVID-19 infection is small and of disparate vintage and clinical features. However, the fact that the spectral patterns in this heterogeneous group of patients congregated in a similar manner actually strengthens their argument that Covid-19 infection results in a distinctive urinary biochemical response.

Minor Issues

None - the paper is extremely well written and includes three figures that are clear and helpful in displaying the data.

Reviewer #2: In the paper entitled “Alterations in the molecular composition of COVID-19 patient urine, detectedusing Raman spectroscopic/computational analysis”, authors propose a new technology to detect COVID-19 disease, identifying a peculiar urine signature, caused by systemic metabolic, inflammatory, and immunologic reactions to infection. The technology is based on Raman spectroscopy and computational analysis. Urine from patients Covid positive were compared urine from healthy individuals, and with those of patients subjected to peritoneal dialysis patients or patients with active bladder cancer. Results show that Raman Chemometric Urinalysis had an overall prediction accuracy of 93.8% for detecting complex, multimolecular fingerprints in urine associated with COVID-19 disease.

The paper has high novelty and the topic is of interest. This is the first study demonstrating that SARS-CoV-2 infection changes the chemical composition of urine. Further and deeper studies will need to better understand present results. Moreover, this method is of interest not only for Covid detection. It was already validated to distinguish urine from healthy subject from those of CKD patients.

Of importance, authors detailed limitations of the studies.

Minor comments:

I suggest to better explaining the future possibilities of the use of this technology in Covid patients. It is not clear which can be the main advantages of the application of this technologies to a large cohort of patients. Is it the identification of renal involvement? Is it the creation of a score for systemic metabolic, inflammatory, and immunologic reactions?

Do authors image some predictive applications?

It would be of interest to compare present results with the urine fingerprint of patient with other severe viral infection.

6. PLOS authors have the option to publish the peer review history of their article (what does this mean?). If published, this will include your full peer review and any attached files.

Reviewer #1: **Yes: **Allan Sklar

Reviewer #2: No

---

## [Decision Letter · Decision Letter 1]

20 Jun 2022

Alterations in the molecular composition of COVID-19 patient urine, detected using Raman spectroscopic/computational analysis

PONE-D-21-38432R1

Dear Dr. John Lee Robertson

We’re pleased to inform you that your manuscript has been judged scientifically suitable for publication and will be formally accepted for publication once it meets all outstanding technical requirements.

Kind regards,

Benedetta Bussolati, MD, PhD

Academic Editor

PLOS ONE

Additional Editor Comments (optional):

Reviewers' comments:

Reviewer's Responses to Questions

**Comments to the Author**

1. If the authors have adequately addressed your comments raised in a previous round of review and you feel that this manuscript is now acceptable for publication, you may indicate that here to bypass the “Comments to the Author” section, enter your conflict of interest statement in the “Confidential to Editor” section, and submit your "Accept" recommendation.

Reviewer #1: All comments have been addressed

Reviewer #2: All comments have been addressed

2. Is the manuscript technically sound, and do the data support the conclusions?

Reviewer #1: Yes

Reviewer #2: Yes

3. Has the statistical analysis been performed appropriately and rigorously? 

Reviewer #1: Yes

Reviewer #2: N/A

4. Have the authors made all data underlying the findings in their manuscript fully available?

Reviewer #1: Yes

Reviewer #2: Yes

5. Is the manuscript presented in an intelligible fashion and written in standard English?

Reviewer #1: Yes

Reviewer #2: Yes

6. Review Comments to the Author

Reviewer #1: (No Response)

Reviewer #2: You have amended my comments, so I find the manuscript adequate to be published in the current form.

7. PLOS authors have the option to publish the peer review history of their article (what does this mean?). If published, this will include your full peer review and any attached files.

Reviewer #1: No

Reviewer #2: No

---

## [Editor Report · Acceptance letter]

7 Jul 2022

PONE-D-21-38432R1 

Alterations in the molecular composition of COVID-19 patient urine, detected using Raman spectroscopic/computational analysis 

Dear Dr. Robertson:

I'm pleased to inform you that your manuscript has been deemed suitable for publication in PLOS ONE. Congratulations! Your manuscript is now with our production department. 

Kind regards, 

on behalf of

Prof. Benedetta Bussolati 

Academic Editor

PLOS ONE